# An Adaptive Nutcracker Optimization Approach for Distribution of Fresh Agricultural Products with Dynamic Demands

**Daqing Wu** [1,2,†] **, Rong Yan** [1,†] **, Hongtao Jin** [1,*] **and Fengmao Cai** [3]

1    College of Economics and Management, Shanghai Ocean University, Shanghai 201306, China
2    Department of Nanchang Technology, 901 Yingxiong Dadao, Economic and Technological Development Zone, Nanchang 330044, China
3    Department of Electronic and Electrical Engineering, The University of Sheffield, Sheffield S37AZ, UK
*    Correspondence: htjin@shou.edu.cn
†    These authors contributed equally to this work.

**Abstract:** In the operational, strategic and tactical decision-making problems of the agri-food supply chain, the perishable nature of the commodities can represent a particular complexity problem. It is, therefore, appropriate to consider decision support tools that take into account the characteristics of the products, the needs and the requirements of producers, sellers and consumers. This paper presents a green vehicle routing model for fresh agricultural product distribution and designs an adaptive hybrid nutcracker optimization algorithm (AH-NOA) based on k-means clustering to solve the problem. In the process, the AH-NOA uses the CW algorithm to increase population diversity and adds genetic operators and local search operators to enhance the global search ability for nutcracker optimization. Finally, the experimental data show that the proposed approaches effectively avoid local optima, promote population diversity and reduce total costs and carbon emission costs.

**Keywords:** dynamic demand; adaptive nutcracker optimizer algorithm; green vehicle routing problem; fresh agricultural products

## 1. Introduction

In recent years, China has issued many relevant documents on accelerating the high-quality development of cold chain logistics and transportation. It has led to the great development of the fresh agricultural products supply chain. More people are choosing to buy fresh agricultural products through e-commerce. On the one hand, fresh agricultural products are perishable, with a short shelf life, high losses and require temperature control. On the other hand, during the cold chain logistics delivery process, fuel consumption and carbon emissions are much higher than in normal logistics. The harm to the environment is higher. Therefore, the study of the fresh dynamic vehicle routing problem will be significant in reducing the commodity value loss, achieving green logistics and sustainable economic development.

This paper investigates the green capacitated vehicle routing problem with dynamic demand (GCVRPDD), which encompasses four topical problems in current vehicle routing research: the vehicle routing problem in cold chain logistics distribution (VRPCLD), the green vehicle routing problem (GVRP), the dynamic vehicle routing problem (DVRP) and the capacitated vehicle routing problem (CVRP). For VRPCLD, Hsu et al. [1] studied the optimal food delivery cycle under multi-vehicle delivery and multi-temperature delivery. Gharehyakheh et al. [2] combined the path optimization problem with temperature, commodity shelf life and energy consumption prediction models. Wu [3] studied the time-dependent split delivery green vehicle routing problem with multiple time windows (TDSDGVRPMTW). Zhang [4] started with a minimized delivery cost function to obtain

a more satisfying cold chain delivery. From the above, it can be found that the multi-temperature distribution of fresh agricultural products provides a methodological entry point for research on cold chain distribution.

For the GVRP, Zhou et al. [5] provided a review of models and solution algorithms for green vehicle paths. Zhang et al. [6] compared the effect of considering carbon emissions on path optimization in solving DVRP. Elgharably et al. [7] proposed a stochastic GVRP under the condition that economic, environmental and social aspects are considered simultaneously. Bruglieri et al. [8], based on new energy vehicles, studied the vehicle routing problem in the existence of gas stations. Zhou et al. [9], Li and Zhang [10] and Cai et al. [11] took the optimization of speed into account when studying the carbon emission pollution problem. Yin et al. [12] took the carbon emission allowance and trading policy as the research object and built an upper-layer carbon trading benefit model from the government's perspective and a lower-layer model from the company's perspective to describe the vehicle routing problem based on the Stackelberg game framework. The above research literature provides the theoretical basis for the calculation of carbon emission costs in this paper.

For the DVRP, Yang et al. [13] proposed a dynamic optimization strategy based on linear programming theory to cope with customer demand changes. Zhang et al. [14] analyzed the dynamic events and transformed the dynamic problem into a static problem. Guo et al. [15] studied the dynamic carpooling problem based on the school buses dynamically carrying students and solved it based on the decomposition of the shared network concepts. Pan et al. [16] proposed a deep reinforcement learning framework to meet the uncertain customer service demands and the training path planning process dynamics. From the above, the majority of research on the green cold chain vehicle path problem has focused on the static vehicle path problem, without considering the customers' dynamic demands in fresh agricultural products distribution.

For the CVRP, Kucuk et al. [17] presented constraint programming-based solution approaches for the three-dimensional loading CVRP. Aydinalp Birecik et al. [18] presented an interactive fuzzy approach for solving green CVRP with imprecise travel time for each vehicle and supplier demand. Wang et al. [19] proposed a novel genetic programming approach to simplify the routing policies. Souza et al. [20] proposed a hybrid algorithm based on a discrete adaptation of the differential evolution meta-heuristic, which is designed for continuous problems, combined with local search procedures to solve the CVRP. The above literature provides many excellent algorithms for solving the CVRP.

Vehicle routing problems are difficult to solve using exact mathematical analytical methods. While heuristic algorithms have good results in solving vehicle routing optimization problems, such as ant colony algorithms [21–23], genetic algorithms [24–26], and particle swarm algorithms [27,28]. Genetic algorithms have disadvantages such as large computational effort and slow convergence, and heuristic algorithms have the disadvantage of large influencing parameters. The nutcracker optimizer algorithm (NOA) [29] has the advantages of high accuracy, fast calculation speed and few parameters. However, there are disadvantages such as easily falling into local optimum and slow convergence speed. To cope with these disadvantages, this paper investigates the impact of carbon emissions on dynamic vehicle path optimization based on temperature control. Based on temperature and geographical location, customers are clustered. To establish a distribution optimization model to minimize the total cost, which contains the vehicle dispatch cost, vehicle transportation cost, value loss, temperature control cost and carbon emission cost, this paper designs an adaptive hybrid nutcracker optimization algorithm (AH-NOA) based on k-means clustering to plan pathways and provides a theoretical basis for logistics companies.

The innovations and contributions of this paper are described as follows:

- An adaptive hybrid nutcracker optimization algorithm combining genetic algorithm and local search operation is designed, which considers both the search breadth and depth. The population is disturbed by the crossover and mutation operation of the

genetic algorithm, and the excellent nutcrackers in the population are deeply searched by the local search operation.

- The GCVRPDD model has been applied rarely in the research of fresh commodity distribution. To make the model better simulate the actual conditions of fresh commodity distribution, this study will consider and evaluate dynamic demand, carbon emissions and temperature control on the basis of GCVRPDD.
- In order to improve the quality and diversity of the initial population, two different methods were used to generate the initial population in this paper. The two methods are, respectively, the CW saving algorithm and the Random method.

The remainder of this paper is organized as follows: Section 2 briefly introduces the problem and constructs the GCVRPDD model. Section 3 presents a new solution algorithm. In Section 4, the results of the experiments are analyzed. Finally, the conclusion is given in Section 5.

## 2. Problem Description and Model Construction

### 2.1. Problem Description

The GCVRPDD proposed in this paper can be described as follows: the cold chain distribution vehicles depart from the distribution center visit customer nodes in the order of the distribution scheme, update the distribution route periodically when the customer's demand is dynamically adjusted and finally return to the distribution center. Suppose $\{0\} \cup V_c$ denotes the set of nodes and $V_c$ denotes the set of customer points before starting distribution. At this time, the customer coordinates and the goods demand $q_{im}^w$ is known. $K = \{1, 2, 3, \cdots, k\}$ is the set of vehicles and the models are the same. The operating hours of the distribution center are $\left[T_s, T_f\right]$. The vehicle does not stay at the customer's location during the delivery. The road conditions are stable and fluctuations in vehicle speeds can be ignored. The dispatch center can hold all the information and regulate the vehicles in real time. At the same time, the delivery time does not exceed $T_k$ in order to ensure the goods are in good condition.

### 2.2. Vehicle Fuel Consumption Model

Demir et al. [30], Suzuki [31] and Hickman [32] have studied the influence of vehicle fuel consumption and carbon emissions. It is pointed out that the distance traveled plays a major role in fuel consumption and carbon emissions. In addition, factors such as vehicle speed, load and characteristics also have a significant impact. At the same time, various fuel consumption and carbon emission measurement models are proposed. In this paper, the MEET model proposed by Hickman is used to complete the calculation. The MEET model includes a carbon emission rate estimation function, a load correction factor and a road slope correction factor. This is applicable to heavy goods vehicles in the weight range of 3.5–32 t. The carbon emission estimation function is:

$$\varepsilon(v) = \omega_0 + \omega_1 v + \omega_2 v^2 + \omega_3 v^3 + \frac{\omega_4}{v} + \frac{\omega_5}{v^2} + \frac{\omega_6}{v^3} \tag{1}$$

where $\varepsilon(v)$ denotes the carbon emission rate when the vehicle is unloaded and driving on a road with a slope of zero. $v$ denotes the vehicle speed. $\omega_0$, $\omega_1$, $\omega_2$, $\omega_3$, $\omega_4$, $\omega_5$ and $\omega_6$ denotes a predefined parameter, which takes different values for trucks with different loads. The load correction factor ($LC$) is as follows:

$$LC = x_0 + x_1 \gamma + x_2 \gamma^2 + x_3 \gamma^3 + x_4 v + x_5 v^2 + x_6 v^3 + \frac{x_7}{v} \tag{2}$$

where $\gamma$ denotes the ratio between the actual load of the vehicle and its capacity. $\chi_0$, $\chi_1$, $\chi_2$, $\chi_3$, $\chi_4$, $\chi_5$, $\chi_6$ and $\chi_7$ are predefined parameters. The value varies with trucks. The vehicle carbon emission rate is $c = \varepsilon(v) LC / 1000$.

During the whole distribution process, the carbon emissions produced by the unit of fuel consumption usually have a fixed value. If the carbon emission caused by 1 L

of gasoline is 2.3 kg, the fuel consumption generated by 1 kg of carbon emission is $\widehat{o} = 1/2.3 = 0.4348$ L/kg. The vehicle fuel consumption rate is $f = \delta c$. Therefore, the fuel consumption of vehicle $k$ traveling from node $i$ to $j$ is $F_{ijw}^k = f d_{ij}$.

### 2.3. Fresh Agricultural Products Quality Attenuation Model

During transportation, the quality of fresh agricultural products gradually declines. The decline rate is strongly related to the transportation environment, time and transport process stability [33,34].

In this paper, the method discussed in [35] is used to calculate the value loss ($vl_{im}^{wk}$) based on the time length and temperature ($T(K)$).

$$vl_{im}^{wk} = q_{im}^w \left[ 1 - e^{\wedge \partial_m^w t_{i0w}^k} \right] p_m \tag{3}$$

where $q_{im}^w$ denotes the demand of customer $i$ for the fresh products ($m$) at a temperature of $w$. $\partial_m^w$ denotes the freshness factor of a unit fresh product ($m$) for delivery at a temperature of $w$. $p_m$ is the price of the fresh commodity $m$. $t_{i0w}^k$ denotes the delivery time that the vehicle takes to get from the distribution center to the customer ($i$).

### 2.4. Temperature Cost Model

During transport, different fresh agricultural products have different temperature requirements and the delivery vehicle needs to set the appropriate temperature to suit the attributes carried. When the performance index $COP$ is used to calculate the energy value change [36], there is $COP = T_L/(T_H - T_L)$. Where $T_H$ is the absolute outside ambient temperature and $T_L$ is the absolute target temperature after temperature control. The physical quantity corresponding to the absolute temperature is the thermodynamic temperature, denoted as $T(K)$, with the symbol K. $T(K)$ is related to the Celsius temperature ($t$) by the equation $T(K) = 273 + t(°C)$. For example, when $T_H$ is 25 °C (298 K) and $T_L$ is 6 °C (279 K). $COP = 14.68$ means that at an ambient temperature of 25 °C, the vehicle temperature is maintained at 6 and needs the temperature control device to absorb 14.68 units of heat. Assuming a temperature control cost of 1 at this point, the temperature control cost factor ($\theta_{mw}$) of 5 °C is $14.68 \cdot 13.9^{-1} \approx 1.06$. Similarly, the temperature control cost factor ($\theta_{mw}$) for the other temperature conditions can be obtained, as shown in Table 1.

**Table 1.** $\theta_{mw}$ under different temperatures.

| Temperature | $COP$ | $\theta_{mw}$ | Temperature | $COP$ | $\theta_{mw}$ |
|---|---|---|---|---|---|
| 6 °C | 14.68 | 1 | −3 °C | 9.64 | 1.52 |
| 5 °C | 13.9 | 1.06 | −4 °C | 9.28 | 1.58 |
| 4 °C | 13.19 | 1.11 | −5 °C | 8.93 | 1.64 |
| 3 °C | 12.55 | 1.17 | −8 °C | 8.03 | 1.83 |
| 2 °C | 11.96 | 1.23 | −9 °C | 7.76 | 1.89 |
| 1 °C | 11.42 | 1.29 | −10 °C | 7.51 | 1.95 |
| 0 °C | 10.92 | 1.34 | −11 °C | 7.28 | 2.02 |
| −1 °C | 10.46 | 1.40 | −12 °C | 7.05 | 2.08 |
| −2 °C | 10.04 | 1.46 | −13 °C | 6.84 | 2.15 |

### 2.5. Green Capacitated Vehicle Routing Problem with Dynamic Demand Model

In this paper, a combination of the initial distribution path and dynamic adjustment strategy is used to solve GCVRPDD. The concept is shown in Figure 1.

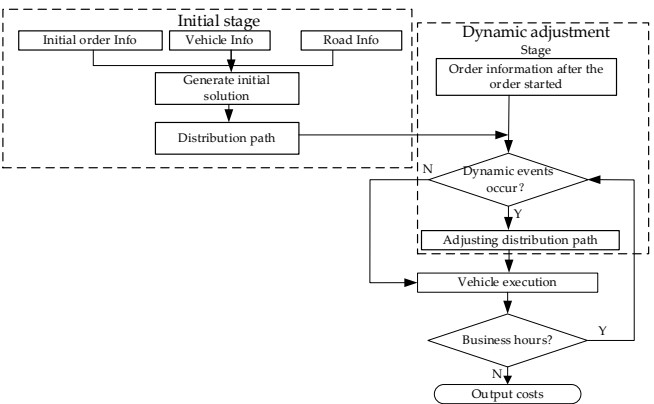

**Figure 1.** Initial planning + dynamic adjustment strategy.

2.5.1. Initial Stage

(1)　Symbol Description

Based on the needs of building the model, this paper uses the corresponding symbols which are listed in Table 2.

**Table 2.** Symbol definition in the GCRPDD optimization.

| Parameter | Definition | Parameter | Definition |
|---|---|---|---|
| $C_1$ | The vehicle dispatch cost | $st_0^k$ | Time of vehicle departure from distribution center |
| $C_2$ | The transportation cost per distance | $T_k$ | The maximum vehicle time in transit |
| $C_3$ | The fuel price | $W$ | A set of temperature control ranges |
| $C_4$ | The unit price of carbon emissions | $F_1$ | The delivery vehicle dispatch cost |
| $Cap$ | The rated load capacity of the vehicle | $F_2$ | The transport cost |
| $TS_i$ | The customer service time | $F_3$ | The temperature control cost |
| $d_{ij}$ | The distance between $i$ and $j$ | $F_4$ | The temperature control cost |
| $t^k_{ijw}$ | The travel time of vehicle $k$ from $i$ to $j$ at temperature $w$ | $F_5$ | The carbon emission cost |
| $x^{wk}_{ij}$ | 0–1 decision variable,1 if vehicle $k$ travels from $i$ to $j$ at temperature $w$ and 0 otherwise | $u^k_{mw} = C_3 f_{kw}\theta_{mw}$ | The temperature-controlled cost of fresh commodity $m$ at temperature $w$ |
| $Q^k_{ijw}$ | The load capacity of vehicle $k$ from customer $i$ to $j$ at $w$ | $f_{kw}$ | The fuel consumption of a hundred kilometers of vehicle $k$ at temperature $w$ |
| $a$ | The carbon emissions per hour of temperature-controlled equipment | $\theta_{mw}$ | The temperature-controlled cost factor unit of fresh commodity at $w$ |
| $V^a_q$ | The new set of customer | $T^{ak}_s$ | The vehicle's departure time at the distribution center |
| $V^a_c$ | The original set of orders not served constitutes | $VD$ | A set of virtual distribution center |
| $z_k$ | 0–1 decision variable, the newly dispatched vehicle | $Q^k$ | The remaining goods of the vehicle $k$ |
| $K^a$ | The set of vehicles currently performing the delivery task | $K^{a^c}$ | The set of new vehicles to be dispatched ($K^{a^c}$ is the complement of $K^a$) |
| $T^a_{q_i}$ | The change moment in customer demand | $st^k_d$ | The time when the vehicle leaves the distribution center or virtual distribution center. |
| $T_{end}$ | The order deadline | | |

(2)    Initial Stage Vehicle Path Model

Objective function:

$$Z = \min\{F_1 + F_2 + F_3 + F_4 + F_5\} \tag{4}$$

$$F_1 = C_1 \sum_{j \in V_c} \sum_{k \in K} x_{0j}^{wk} \tag{5}$$

$$F_2 = C_2 \sum_{w \in W} \sum_{j \in \{0\} \cup V_c} \sum_{i \in \{0\} \cup V_c} \sum_{k \in K} F_{ijw}^k x_{ij}^{wk} \tag{6}$$

$$F_3 = \sum_{k \in K} \sum_{m \in M} \sum_{w \in W} \sum_{j \in \{0\} \cup V_c} \sum_{i \in \{0\} \cup V_c} t_{ijw}^k u_{mw}^k Q_{ijw}^k \tag{7}$$

To simplify the calculation, it is assumed that no temperature control cost is generated in return.

$$F_4 = \sum_{i \in V_c} \sum_{m \in M} \sum_{w \in W} \sum_{k \in K} v l_{im}^{wk} \tag{8}$$

$$F_5 = C_4 \sum_{w \in W} \sum_{j \in \{0\} \cup V_c} \sum_{i \in \{0\} \cup V_c} \sum_{k \in K} \left( F_{ijw}^k x_{ij}^{wk} / \widehat{o} + a t_{ijw}^k x_{ij}^{wk} \right) \tag{9}$$

s.t.

$$\sum_{i \in \{0\} \cup V_c} \sum_{j \in V_c} \sum_{w \in W} x_{ij}^{wk} q_{jm}^w \le Cap, \forall k \in K \tag{10}$$

$$\sum_{j \in \{0\} \cup V_c} \sum_{k \in K} \sum_{w \in W} x_{ij}^{wk} = 1, \forall i \in V_c \tag{11}$$

$$\sum_{i \in \{0\} \cup V_c} \sum_{k \in K} \sum_{w \in W} x_{ij}^{wk} = 1, \forall j \in V_c \tag{12}$$

$$\sum_{i \in \{0\} \cup V_c} \sum_{w \in W} x_{ij}^{wk} = \sum_{i \in \{0\} \cup V_c} \sum_{w \in W} x_{ji}^{wk}, \forall j \in V_c, \forall k \in K \tag{13}$$

$$\sum_{j \in V_c} \sum_{w \in W} x_{0j}^{wk} = \sum_{i \in V_c} \sum_{w \in W} x_{0i}^{wk} \le 1, \forall k \in K \tag{14}$$

$$T_s + \sum_{i \in \{0\} \cup V_c} \sum_{j \in \{0\} \cup V_c} \sum_{w \in W} t_{ijw}^k x_{ij}^{wk} + \sum_{i \in \{0\} \cup V_c} \sum_{j \in V_c} \sum_{w \in W} q_{jm}^w TS_j x_{ij}^{wk} \le T_f, \forall k \in K \tag{15}$$

$$\sum_{i \in \{0\} \cup V_c} \sum_{j \in V_c} \sum_{w \in W} d_{ij} x_{ij}^{wk} \le v T_k, \forall k \in K \tag{16}$$

$$\sum_{i \in S} \sum_{j \in S} \sum_{w \in W} x_{ij}^{wk} \le |S| - 1, S \subseteq V_c, S \ne \varnothing, \forall k \in K \tag{17}$$

$$\sum_{i,j \in V_c} \sum_{k \in K} \sum_{w \in W} x_{ij}^{wk} \le |K|, \forall k \in K \tag{18}$$

$$x_{ij}^{wk} \in \{0,1\}, \forall i, j \in \{0\} \cup V_c, \forall k \in K, w \in W \tag{19}$$

Equation (10) indicates that the sum of the orders does not exceed the vehicle load limit. Equations (11) and (12) indicate that the customer point is served by a vehicle once and only once. Equation (13) indicates that the vehicle must leave after completing the order. Equation (14) represents each vehicle serving only one route, and the vehicle departs from the distribution center and returns after completing the order. Equation (15) represents the vehicle returns to the distribution center at a time no later than the operating deadline. Equation (16) represents the delivery time constraint. Equation (17) represents the

elimination of the sub-circuit constraint. Equation (18) represents the number of vehicles performing distribution tasks that do not exceed the maximum number in the distribution center. Equation (19) is the decision variable.

2.5.2. Dynamics Adjustment Stage

(1)　Dynamic Adjustment Strategy

The dynamic character of GCVRPDD is mainly reflected in customer order updates, where the demand changes in real time during the distribution. In the dynamic adjustment stage, this paper divides the operation time into several homogeneous time slices and adopts a periodic optimization strategy for optimization. Dynamic events that occur in the current time slice are not processed immediately but are globally optimized in the next time slice together with customers who have not received service in the path. With demand dynamically updated, the VRPs within each time slice are static, and the optimization solutions *S* are continuously passed between time slices. As a result, GCVRPDD is converted into a number of correlated static vehicle routing problems (SVRP), as shown in Figure 2. To simplify the calculation, it is set that the customer being delivered cannot change the demand.

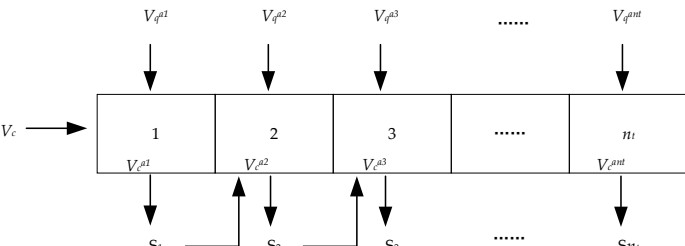

**Figure 2.** GCVRPDD rolling time domain dynamic optimization.

The number of time slices is set to $n_t$, and each time slice is $\left(T_f - T_s\right)/n_t$ [37]. The dynamic demand in the current time slice is not processed immediately and will be passed to the next time slice for optimization. With a larger $n_t$, the dynamic demand response is faster. However, a larger $n_t$ does not necessarily result in a better pathway and also leads to system redundancy in computational space [38].

(2)　Dynamic Adjustment Model

In this paper, according to the dynamic event occurrence time, the vehicle location is determined and set as a virtual distribution point. The objective is to minimize the cost and value loss to complete subsequent distribution tasks after a dynamic event has occurred. During the optimization process, the position of vehicle $k$ and the goods surplus ($Q^k$) in the dynamic adjustment phase can be determined from the previous phase. According to parameters, such as vehicle speed and load, the cost and value loss of subsequent distribution tasks are solved. Suppose the vehicle's ($k$) initial distribution path is 0-1-2-3-4-0. In dynamic adjustment, if the vehicle is at customer point 2, the reconfiguration takes customer point 2 as the virtual distribution center. At this point, the goods remaining of vehicle $k$ is $Q^k = Cap - q_{1m}^w - q_{2m}^w$. If the vehicle is between customer points 2 and 3, the reconfiguration takes the vehicle's current location as the virtual point and does not change the delivery task to customer point 3. The goods remaining are $Q^k = Cap - q_{1m}^w - q_{2m}^w - q_{3m}^w$. If the optimized subsequent distribution path is 3-5-6-0, the distribution path for vehicle $k$ is 0-1-2-3-5-6-0, based on the previous path. The model is as follows:

$$Z = \min\{F_1 + F_2 + F_3 + F_4 + F_5\} \tag{20}$$

$$F_1 = C_1 \sum_{j \in V_c^a} \sum_{k \in K^{a^c}} \left(x_{0j}^{wk} - z^k\right) \tag{21}$$

$$F_2 = C_2 \sum_{w \in W} \sum_{j \in \{0\} \cup V_c} \sum_{i \in \{0\} \cup V_c} \sum_{k \in K} F_{ijw}^k x_{ij}^{wk} \tag{22}$$

$$F_3 = \sum_{k \in K} \sum_{m \in M} \sum_{w \in W} \sum_{j \in \{0\} \cup V_c^a \cup VD} \sum_{i \in \{0\} \cup V_c^a \cup VD} t_{ijw}^k u_{mw}^k Q_{ijw}^k \tag{23}$$

$$F_4 = \sum_{i \in \{0\} \cup V_c^a \cup VD} \sum_{m \in M} \sum_{w \in W} \sum_{k \in K} v l_{im}^{wk} \tag{24}$$

$$F_5 = C_4 \sum_{w \in W} \sum_{j \in \{0\} \cup V_c} \sum_{i \in \{0\} \cup V_c} \sum_{k \in K} \left( F_{ijw}^k x_{ij}^{wk} / \widehat{o} + a t_{ijw}^k x_{ij}^{wk} \right) \tag{25}$$

s.t.

$$\sum_{i \in VD \cup V_c^a} \sum_{j \in V_c^a} \sum_{w \in W} x_{ij}^{wk} q_{jm}^w \leq Q^k, \forall k \in K^a \tag{26}$$

$$\sum_{i \in \{0\} \cup V_c^a} \sum_{j \in V_c^a} \sum_{w \in W} x_{ij}^{wk} q_{jm}^w \leq Cap, \forall k \in K^{a^c} \tag{27}$$

$$|K^a| + \sum_{j \in V_c^a} \sum_{k \in K^{a^c}} \sum_{w \in Q} x_{0j}^{wk} \leq |K| \tag{28}$$

$$\sum_{j \in \{0\} \cup V_c^a} \sum_{k \in K} \sum_{w \in W} x_{ij}^{wk} = 1, \forall i \in V_c^a \tag{29}$$

$$\sum_{i \in \{0\} \cup V_c^a} \sum_{k \in K} \sum_{w \in W} x_{ij}^k = 1, \forall j \in V_c^a \tag{30}$$

$$\sum_{i \in \{0\} \cup V_c^a \cup VD} \sum_{w \in W} x_{ij}^{wk} = \sum_{i \in \{0\} \cup V_c^a} \sum_{w \in W} x_{ji}^{wk}, \forall j \in V_c^a, \forall k \in K \tag{31}$$

$$\sum_{j \in V_c^a} \sum_{w \in W} x_{0j}^{wk} = \sum_{i \in V_c^a} \sum_{w \in W} x_{i0}^{wk} \leq 1, \forall k \in K^{a^c} \tag{32}$$

$$\sum_{k \in K^a} \sum_{j \in V_c^a} \sum_{w \in W} x_{ij}^{wk} = 1, \forall i \in VD \tag{33}$$

$$\sum_{k \in K^a} \sum_{i \in \{0\} \cup V_c^a} \sum_{w \in W} x_{ij}^{wk} = 0, \forall i \in VD \tag{34}$$

$$\sum_{k \in K} \sum_{i \in VD \cup V_c^a} \sum_{w \in W} x_{i0}^{wk} = \sum_{k \in K^{a^c}} \sum_{i \in V_c^a} \sum_{w \in W} x_{0j}^{wk} + |K^a| \tag{35}$$

$$T_{q_i}^a \leq T_{end}, \forall i \in V_q^a \tag{36}$$

$$T_s^{ak} + \sum_{i \in \{0\} \cup V_c^a \cup VD} \sum_{j \in \{0\} \cup V_c^a} \sum_{w \in W} t_{ijw}^k x_{ij}^{wk} + \sum_{i \in \{0\} \cup V_c^a \cup VD} \sum_{j \in V_c^a} \sum_{w \in W} q_{jm}^w T S_j x_{ij}^{wk} \leq T_f, \forall k \in K \tag{37}$$

$$\sum_{i \in S} \sum_{j \in S} \sum_{w \in W} x_{ij}^{wk} \leq |S| - 1, |S| = \sum_{j \in V_c^a} \sum_{w \in W} x_{ij}^{wk}, \forall k \in K \tag{38}$$

$$\sum_{i \in \{0\} \cup V_c} \sum_{j \in V_c} \sum_{w \in W} d_{ij} x_{ij}^{wk} \leq v T_k, \forall k \in K \tag{39}$$

$$x_{ij}^{wk} \in \{0, 1\}, \forall i, j \in V_c^a, \forall k \in K, w \in W \tag{40}$$

$$z^k \in \{0, 1\}, \forall k \in K^{a^c} \tag{41}$$

Equations (26) and (27) represent the vehicle load constraint. Equation (28) is the number of delivery vehicles constraint. Equations (29) and (30) denote that the customer has and will only be served once. Equation (31) means that the vehicle leaves the customer's point after service. Equation (32) indicates that each vehicle serves only one path and that the vehicle departs from the distribution center and returns to the distribution center after completing the order. Equations (33) and (34) mean that only one vehicle leaves the virtual distribution point and does not return. Equation (35) indicates that all vehicles performing a distribution task return to the distribution center after completing all tasks. Equation (36) indicates that the time to process order changes does not exceed the order deadline of the distribution center. Equation (37) represents the vehicle returns to the distribution center at a time no later than the operating deadline. Equation (38) eliminates the sub-loop constraint. Equation (39) is the delivery time constraint. Equations (40) and (41) are the decision variable attributes.

## 3. Problem Solution and Algorithm Design

### 3.1. Problem Solution

Step 1: Determine the initial distribution path of the vehicles. The vehicles all depart from the distribution center. According to the pre-departure order information, using the AH-NOA based on k-means clustering to determine the initial plan.

Step 2: Determine whether a dynamic event occurs according to the rolling time domain. If it does, skip to Step 3, otherwise, end.

Step 3: According to the event occurrence time, the vehicle driving path before the dynamic event occurs is derived.

Step 4: Dynamic events occur and relevant order information is updated.

Step 5: Problem transformation. Set up a virtual distribution center based on the location information of vehicles in transit. Integrate undelivered orders and new orders into a new order requirement. Transforming the single-center DVRP into a new multi-center SVRP.

Step 6: Determine the vehicle path after the dynamic event occurs. Based on the new problem obtained in Step 5, the AH-NOA is used to solve it. Use the vehicle path obtained in Step 3 to replace the path between the distribution center and the virtual distribution center. Skip to Step 2.

### 3.2. The AH-NOA Design Based on K-Means Clustering Algorithm

#### 3.2.1. K-Means Clustering Algorithm

The k-means clustering analysis of customers can reduce the decay of commodity quality and increase customer satisfaction. Because of the integrated consideration of the commodity's temperature and customer location, it can make the solution more realistic.

Suppose the location coordinates of the customer ($i$) and demand temperature interval of fresh agricultural products are $(x_i, y_i)$,$(lp_i, up_i)$, respectively. The clustering distance between customers $i$ and $j$ is $d_{ij}^c = |x_i - x_j| + |y_i - y_j| + tr|lp_i - lp_j| + tr|up_i - up_j|$ [39]. $tr$ denotes the conversion factor between distance and temperature. The specific steps are as follows:

(1)  Import customer location, commodity temperature and $tr$.
(2)  The products are divided into several temperature intervals according to their temperature characteristics while taking into account the customer's freshness demands.
(3)  The initial number of clustering units is determined as $qc$ based on the k-means clustering algorithm, and the $qc$ initial clustering centers are selected in each temperature control interval.
(4)  Under different temperature intervals, calculate the distance from the customer point to the cluster center and assign the clustered customers to the closest clustered unit.
(5)  Add the number of clustering units as $qc = qc + 1$ and generate new clustering units and centers.
(6)  Repeat (4) and (5) until the cluster centers no longer change.

### 3.2.2. Nutcracker Optimizer Algorithm

The nutcracker optimizer algorithm is a swarm intelligence optimization algorithm proposed in 2023. In the summer and autumn, nutcrackers search for food at random and choose to transport the better food to a storage area away from the foraging area. In winter and spring, nutcrackers go to storage areas to find food, which may not necessarily be their food but may also become the food of others. NOA simulates the foraging and food storage behavior based on the nutcracker's ability to find food by relying on spatial memory mechanisms and achieves the goal of finding the best food.

Heuristic algorithms have good results for solving vehicle routing problems. However, it has a lack of robustness, and the results are easily influenced by a larger number of parameters. NOA has the advantages of high accuracy, fast computation and a small number of parameters. Moreover, due to the characteristics of the problem studied in this paper, a fast computational algorithm is more beneficial to solve the problem. However, there are also disadvantages such as the tendency to fall into local optima and slow convergence. The flow of this algorithm is shown in Figure 3.

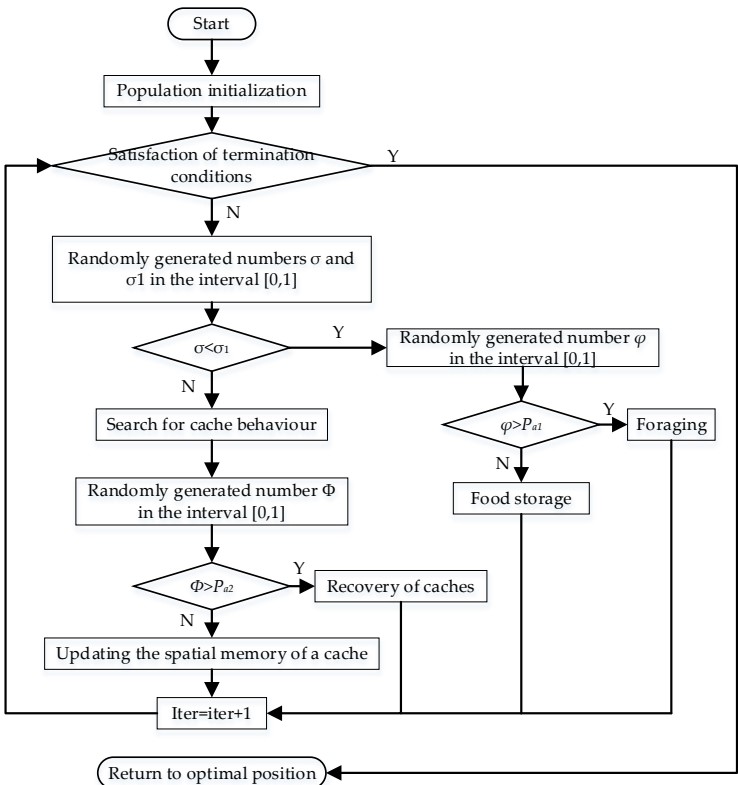

**Figure 3.** NOA flow chart.

### 3.2.3. Adaptive Hybrid Nutcracker Algorithm

To address the disadvantages of the NOA, an adaptive hybrid NOA is proposed. To ensure the diversity of the population, the CW algorithm was used to initialize some of the solutions. To keep the overall quality of the population, better nutcrackers are selected to generate new nutcrackers through the genetic operators and local search operators. A nutcracker winter food search probability and a local optimization avoidance probability are introduced into the decline function so that it can be adaptively adjusted with iterations. In this way, the algorithm's convergence speed can be increased. It is possible to avoid falling into a local optimum to a certain extent. The AH-NOA flow chart is shown in Figure 4.

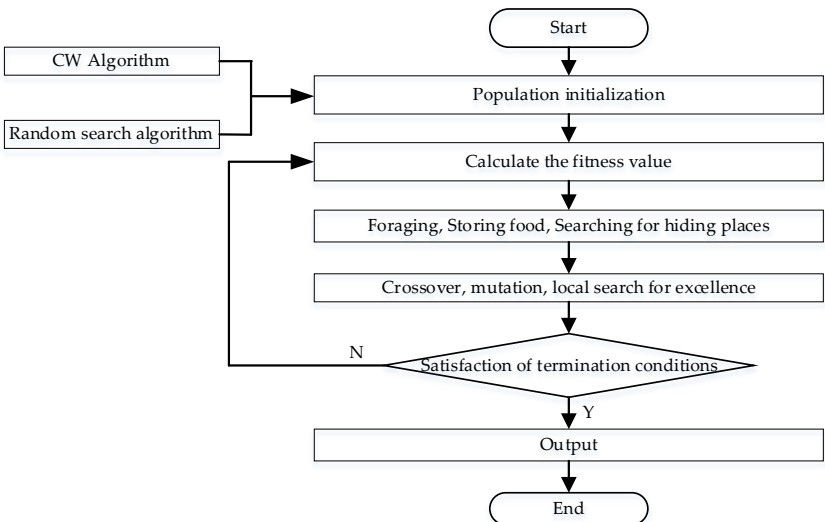

**Figure 4.** AH-NOA flow chart.

The specific steps of the AH-NOA are as follows:

(1)  Initialize the population. To increase the diversity of solutions, the initial population is generated in two ways: CW algorithm and random search.

(2)  Adaptability function. The total cost is used as the fitness function of the algorithm

(3)  Parameter iteration. $Pa_1$ controls the search direction in the storage phase. $Pa_2$ determines the swapping probability between the cache search and recovery phases. $\delta$ avoids getting stuck in a local optimum. The $Pa_1$, $Pa_2$ and $\delta$ are improved to adaptively change with the algorithm running using the following equations:

$$Pa_1 = (iter\_max - iter)/iter\_max \tag{42}$$

$$Pa_2 = 1/(1 - 1.5(\log_2(1/iter))) \tag{43}$$

$$\delta = {}^{iter\_max-iter+1}\sqrt{\exp(-iter^4/iter\_max^3)}0.05 \tag{44}$$

(4)  Movement strategy. Each nutcracker performed foraging, storing food, searching for caches and obtaining food behaviors according to Equations (45)–(51), respectively, based on its current food search status. $\overrightarrow{X}_k^{iter+1}$ denotes the position of the $k$ nutcracker in the *iter* generation. $U_j$ and $L_j$ denote the upper and lower bounds of the nutcracker's position at the point $j$. $\gamma$ is a random number generated from a levy flight. $X_{Mj}^{iter}$ denotes the average position of all nutcrackers at point $j$ in the *iter* generation. $A, B$ and $C$ are three nutcrackers randomly selected from the population to facilitate the search for high-quality food sources. $\tau_1$, $\tau_2$, $r$ and $r_1$ are real numbers randomly generated in the interval $[0, 1]$. $\mu$ is randomly generated in the interval $[0, 1]$ based on a normal distribution.

$$\overrightarrow{X}_k^{iter+1} = \begin{cases} X_{kj}^{iter} & \tau_1 < \tau_2 \\ X_{Mj}^{iter} + \gamma\left(X_{Aj}^{iter} - X_{Bj}^{iter}\right) + \mu\left(r^2 U_j - L_j\right) & \tau_1 \geq \tau_2, iter < iter\_max/2 \\ X_{Cj}^{iter} + \mu\left(X_{Aj}^{iter} - X_{Bj}^{iter}\right) + \mu(r_1 < \delta)\left(r^2 U_j - L_j\right) & otherwise \end{cases} \tag{45}$$

$$\vec{X}_k^{iter+1} = \begin{cases} \vec{X}_k^{iter} + \mu\left(\vec{X}_{best}^{iter} - \vec{X}_k^{iter}\right)|\lambda| + r_1\left(\vec{X}_A^{iter} - \vec{X}_B^{iter}\right) & \tau_1 < \tau_2 \\ \vec{X}_{best}^{iter} + \mu\left(\vec{X}_A^{iter} - \vec{X}_B^{iter}\right) & \tau_1 < \tau_3 \\ \vec{X}_{best}^{iter}l & otherwise \end{cases} \tag{46}$$

$$\vec{RP}_{k,1}^{iter} = \begin{cases} \vec{X}_k^{iter} + \alpha\cos(\theta)\left(\vec{X}_A^{iter} - \vec{X}_B^{iter}\right) + \alpha RP & \theta = \pi/2 \\ \vec{X}_k^{iter} + \alpha\cos(\theta)\left(\vec{X}_A^{iter} - \vec{X}_B^{iter}\right) & otherwise \end{cases} \tag{47}$$

$$\vec{RP}_{k,2}^{iter} = \begin{cases} \vec{X}_k^{iter} + \left(\alpha\cos(\theta)\left(\left(\vec{U} - \vec{L}\right)\tau_3 + \vec{L}\right) + \alpha RP\right)\vec{U}_2 & \theta = \pi/2 \\ \vec{X}_k^{iter} + \alpha\cos(\theta)\left(\left(\vec{U} - \vec{L}\right)\tau_3 + \vec{L}\right)\vec{U}_2 & otherwise \end{cases} \tag{48}$$

$\lambda$ is a random number generated from a levy flight. $X_{best}^{iter}$ is the position of the current optimal nutcracker. $\tau_3$ is a randomly generated real number in the interval $[0,1]$. $l$ is the diversity factor of the solution linearly decreasing from 1 to 0. $\left(\vec{RP}_{k,1}^{iter}, \vec{RP}_{k,2}^{iter}\right)$ denotes the location of the $k$ nutcracker storing food in generation $iter$. $\vec{U} = (U_1, U_2, \ldots, U_j)$ and $\vec{L} = (L_1, L_2, \ldots, L_j)$ denote the range of regions where the nutcracker stores food. $RP$ is a random location. $\alpha$ avoids the algorithm falling into a local optimum and adjusts according to Equation (49):

$$\alpha = \begin{cases} (1 - iter/iter\_max)^{2iter/iter\_max} & r_1 > r_2 \\ (iter/iter\_max)^{2/iter} & otherwise \end{cases} \tag{49}$$

$$\vec{X}_k^{iter+1} \begin{cases} \vec{X}_k^{iter} & \tau_7 < \tau_8, \tau_3 < \tau_4 \, or \, \tau_5 < \tau_6 \\ \vec{X}_k^{iter} + r_1\left(\vec{X}_{best}^{iter} - \vec{X}_k^{iter}\right) + r_2\left(\vec{RP}_{k,1}^{iter} - \vec{X}_C^{iter}\right) & \tau_7 < \tau_8, \tau_3 \geq \tau_4 \\ \vec{X}_k^{iter} + r_1\left(\vec{X}_{best}^{iter} - \vec{X}_k^{iter}\right) + r_2\left(\vec{RP}_{k,2}^{iter} - \vec{X}_C^{iter}\right) & \tau_7 < \tau_8, \tau_5 \geq \tau_6 \end{cases} \tag{50}$$

$$\vec{X}_k^{iter+1} = f^{-1}\left(\min\left(f\left(\vec{X}_k^{iter}\right), f\left(\vec{RP}_{k,1}^{iter}\right), f\left(\vec{RP}_{k,2}^{iter}\right)\right)\right) \tag{51}$$

$r_2$, $\tau_4$, $\tau_5$, $\tau_6$, $\tau_7$ and $\tau_8$ are a randomly generated real number in the interval $[0,1]$.

Suppose a vehicle departs from distribution center 0 and needs to make a delivery to customer points 1 and 2. $X_1^{iter}$ is the position of the current nutcracker 1 and $X_2^{iter}$ is the position of the current nutcracker 2. If $X_1^{iter} \leq X_2^{iter}$, $x_{12}^{wk} = 1$, $x_{21}^{wk} = 0$ and the order of vehicle visits is 0 -1-2-0. Otherwise, $x_{12}^{wk} = 0$, $x_{21}^{wk} = 1$ and the order of vehicle visits is 0-2-1-0.

(5) Crossover, mutation and local search behavior. Incorporating the crossover and mutation operators from the genetic algorithm into the NOA can effectively overcome the NOA's lack of global search capability. The key to enhancing the global search ability is to enhance the diversity of the population, which both genetic operators can achieve. Therefore, the genetic operator is embedded in the NOA to expand the solution space and improve the global search capability.

① Crossover operations

Multiple crossovers can avoid individuals from converging prematurely, but too many crossovers will destroy some better solutions. In this paper, according to the roulette strategy, two crosses, *OA* and *OB*, are selected. After the two individuals have been crossed,

the recurring points are deleted, resulting in two new nutcrackers. The process is shown in Figure 5.

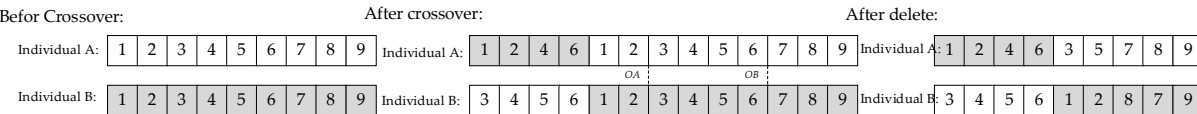

**Figure 5.** Crossover operation.

② Mutation operation

Select two random positions and directly exchange them to complete the mutation operation, as shown in Figure 6.

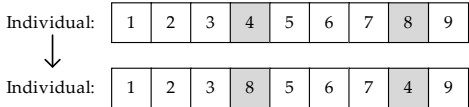

**Figure 6.** Mutation operation.

③ Local search operations

Remove some similar location points based on similarity by using the destruction operator. Under load capacity and time constraints, removed points are inserted back into the total cost. The similarity is calculated as $R_{ij} = 1/(S_{ij} + d_{ij}/d_{ij\_max})$, where $S_{ij} = 1$ means that $i$ and $j$ are not on the same path and $S_{ij} = 0$ means that $i$ and $j$ are on the same path, $d_{ij\_max} = \max\{j \in V_c, i \neq j | d_{ij}\}$.

(6)  Algorithm termination. The algorithm is terminated when the maximum number of iterations has been reached.

### 3.2.4. Time Complexity Analysis of AH-NOA

In the base NOA, if the number of populations is $n$, then when simulating the foraging and storage behavior, the time magnitude is $n$. In the cache-search and recovery strategy, generating the $RP$ matrix needs at most n times and the time magnitude is $n$. Each nutcracker has to search for the best food at most $iter\_max$ times. The time magnitude is $iter\_max$. In summary, the time complexity of the NOA is $O((n+n) \cdot iter\_max)$. AH-NOA adds crossover, mutation and local search behavior to the NOA, which has a time dimension of $n$. The time complexity of the AH-NOA is $O((n+n) \cdot iter\_max + n)$.

### 3.2.5. Algorithm Effectiveness Verification

To verify the effectiveness of the algorithm improvements, a comparison experiment between the basic NOA and AH-NOA was made. Customer location coordinates and service demands were obtained by improving the Solomon dataset [40]. At the same time, the temperature intervals and prices for fresh agricultural products are added, and 12 examples were selected for analysis and comparison. The main parameters are set as follows: $pop_{size} = 50$, $iter\_max = 200$, $p_c = 0.8$, $p_m = 0.1$, $tr = 1.2$. Due to the high information level of the order, the communication time between the delivery personnel and the customer is greatly reduced. To simplify the calculation, assume $TS_i = 0$. The characteristics of the example data set are shown in Table 3. Each set was calculated 20 times and the best values were chosen, as shown in Table 4. Where $TD$ denotes total cost, $VL$ denotes loss of value, $VN$ denotes the number of vehicles and $CEC$ denotes the carbon emissions cost. $Gap\_c = (TD_{NOA} - TD_{AH-NOA})/TD_{NOA}$, $Gap\_g = (CEC_{NOA} - CEC_{AH-NOA})/CEC_{NOA}$.

**Table 3.** Dataset characteristics.

| Case No. | Customer Numbers | Temperature Control Interval | Price |
|---|---|---|---|
| 1–3 | 40 | (−5 °C)–(0 °C) | 10 |
| | | (1 °C)–(6 °C) | 7 |
| 4–6 | 60 | (−5 °C)–(0 °C) | 10 |
| | | (−13 °C)–(−8 °C) | 15 |
| 7–9 | 80 | (−5 °C)–(0 °C) | 10 |
| | | (1 °C)–(6 °C) | 7 |
| | | (−13 °C)–(−8 °C) | 15 |
| 10–12 | 100 | (−5 °C)–(0 °C) | 10 |
| | | (1 °C)–(6 °C) | 7 |
| | | (−13 °C)–(−8 °C) | 15 |

**Table 4.** Comparison of algorithm solution results.

| No. | Name | AH-NOA | | | | NOA | | | | Gap_c | Gap_g |
|---|---|---|---|---|---|---|---|---|---|---|---|
| | | *TD* | *VL* | *CEC* | *VN* | *TD* | *VL* | *CEC* | *VN* | | |
| 1 | C101 | 964 | 183 | 203 | 5 | 1716 | 205 | 235 | 5 | 43.82% | 13.62% |
| 2 | R101 | 1383 | 192 | 341 | 5 | 1713 | 185 | 428 | 6 | 19.26% | 20.33% |
| 3 | RC101 | 1812 | 245 | 403 | 7 | 2540 | 272 | 571 | 9 | 28.66% | 29.42% |
| 4 | C102 | 1746 | 379 | 371 | 6 | 2150 | 364 | 495 | 8 | 18.79% | 25.05% |
| 5 | R102 | 2226 | 317 | 547 | 8 | 2896 | 270 | 732 | 10 | 23.14% | 25.27% |
| 6 | RC102 | 3308 | 422 | 724 | 10 | 3983 | 357 | 905 | 14 | 16.95% | 20.00% |
| 7 | C103 | 2900 | 398 | 403 | 7 | 3705 | 407 | 561 | 9 | 21.73% | 28.16% |
| 8 | R103 | 3340 | 299 | 486 | 7 | 4178 | 325 | 634 | 8 | 20.06% | 23.34% |
| 9 | RC103 | 4222 | 391 | 632 | 9 | 5637 | 346 | 893 | 13 | 25.10% | 29.23% |
| 10 | C104 | 3756 | 511 | 545 | 9 | 5986 | 559 | 972 | 13 | 37.25% | 43.93% |
| 11 | R104 | 4175 | 420 | 705 | 10 | 4522 | 391 | 860 | 12 | 7.67% | 18.02% |
| 12 | RC104 | 4943 | 446 | 638 | 9 | 6374 | 394 | 921 | 13 | 22.45% | 30.73% |
| | Ave | 2898 | 350 | 500 | 8 | 3783 | 340 | 684 | 10 | 24% | 25.59% |
| | *t*-test | | | | | −6.035 | | −6.81 | −6.325 | | |
| | *p*-value | | | | | 0.000059 | | 0.000019 | 0.000038 | | |

Table 4 shows that the AH-NOA outperforms the NOA in terms of *TD*, *CEC* and *VN*. In the pathway optimization model with the lowest total cost, the AH-NOA obtained better results than the NOA. Total costs were optimized by a maximum of 43.82% and an average of 24%. The carbon emissions costs were optimized by a maximum of 43.93% and an average of 25.59%. The *t*-test and *p*-value analysis showed that the difference between the calculated *TD*, *CEC* and *VN* for the AH-NOA solution and the NOA was significant. Therefore, the AH-NOA is an effective improvement to the NOA.

## 4. Example Analysis

### 4.1. Example Design

Before delivery begins, the distribution center receives orders from 60 customers, which are located in different parts of the city. Each customer has different demands and temperature intervals. The distribution center information is shown in Table 5. The specific customer demands are shown in Table 6. Customer location, demand and commodity temperature attributes are randomly generated using the *rand()* function according to objective reality. $Cap = 200$ kg, $n_t = 32$. The hire cost is $C_1 = 60$ CNY. The average vehicle speed is 50 km/h. Assume a unit fuel emission factor of 2.63 kg/L. Referring to the average carbon emission trading price on the Beijing, Shanghai and Guangdong exchanges on 19 April 2019, set the unit carbon emission price to 0.1 CNY/kg.

**Table 5.** Distribution center information.

| X/km | Y/km | $T_s$ | $T_f$ |
|---|---|---|---|
| 0 | 0 | 8:00 | 16:00 |

**Table 6.** Customer information.

| No. | X/km | Y/km | Demand/kg | Temperature Control Interval | No. | X/km | Y/km | Demand/kg | Temperature Control Interval |
|---|---|---|---|---|---|---|---|---|---|
| 1 | −27 | −25 | 8 | (−5 °C)–(0 °C) | 31 | 8 | 15 | 23 | (−5 °C)–(0 °C) |
| 2 | 31 | 3 | 11 | (1 °C)–(6 °C) | 32 | −18 | −10 | 11 | (1 °C)–(6 °C) |
| 3 | −4 | −29 | 25 | (1 °C)–(6 °C) | 33 | 29 | −24 | 3 | (−5 °C)–(0 °C) |
| 4 | 30 | −18 | 12 | (−13 °C)–(−8 °C) | 34 | 25 | 20 | 16 | (−13 °C)–(−8 °C) |
| 5 | 26 | −1 | 2 | (−13 °C)–(−8 °C) | 35 | 21 | 14 | 8 | (−13 °C)–(−8 °C) |
| 6 | −39 | −26 | 21 | (−13 °C)–(−8 °C) | 36 | −4 | −11 | 11 | (−13 °C)–(−8 °C) |
| 7 | −23 | −23 | 10 | (−5 °C)–(0 °C) | 37 | 34 | −12 | 24 | (1 °C)–(6 °C) |
| 8 | −22 | 2 | 11 | (−13 °C)–(−8 °C) | 38 | 1 | −11 | 19 | (1 °C)–(6 °C) |
| 9 | 3 | −33 | 4 | (−13 °C)–(−8 °C) | 39 | 13 | 8 | 17 | (1 °C)–(6 °C) |
| 10 | 9 | 33 | 23 | (−5 °C)–(0 °C) | 40 | 20 | −27 | 13 | (−5 °C)–(0 °C) |
| 11 | 6 | 12 | 25 | (−5 °C)–(0 °C) | 41 | 31 | −39 | 18 | (1 °C)–(6 °C) |
| 12 | 7 | 1 | 15 | (−13 °C)–(−8 °C) | 42 | −20 | 10 | 8 | (−13 °C)–(−8 °C) |
| 13 | 31 | 0 | 2 | (1 °C)–(6 °C) | 43 | −5 | −36 | 3 | (−13 °C)–(−8 °C) |
| 14 | −9 | −29 | 16 | (−13 °C)–(−8 °C) | 44 | −39 | 17 | 8 | (−13 °C)–(−8 °C) |
| 15 | −27 | 1 | 2 | (−13 °C)–(−8 °C) | 45 | −32 | 3 | 17 | (1 °C)–(6 °C) |
| 16 | 4 | 14 | 17 | (−5 °C)–(0 °C) | 46 | 30 | −4 | 25 | (1 °C)–(6 °C) |
| 17 | −9 | 22 | 19 | (−5 °C)–(0 °C) | 47 | 23 | 28 | 6 | (1 °C)–(6 °C) |
| 18 | −1 | −26 | 22 | (−5 °C)–(0 °C) | 48 | −22 | 5 | 24 | (−13 °C)–(−8 °C) |
| 19 | 34 | 20 | 18 | (1 °C)–(6 °C) | 49 | 21 | −22 | 4 | (−13 °C)–(−8 °C) |
| 20 | 11 | −15 | 24 | (1 °C)–(6 °C) | 50 | 18 | 28 | 25 | (1 °C)–(6 °C) |
| 21 | −32 | −32 | 12 | (−13 °C)–(−8 °C) | 51 | 9 | 16 | 13 | (−5 °C)–(0 °C) |
| 22 | 2 | 34 | 5 | (−13 °C)–(−8 °C) | 52 | −5 | −30 | 15 | (1 °C)–(6 °C) |
| 23 | 10 | 9 | 18 | (1 °C)–(6 °C) | 53 | −9 | −1 | 19 | (−13 °C)–(−8 °C) |
| 24 | −24 | −32 | 22 | (−13 °C)–(−8 °C) | 54 | −29 | 30 | 7 | (1 °C)–(6 °C) |
| 25 | −30 | −18 | 12 | (−5 °C)–(0 °C) | 55 | −38 | −22 | 7 | (−5 °C)–(0 °C) |
| 26 | 9 | −9 | 12 | (−13 °C)–(−8 °C) | 56 | −16 | −37 | 13 | (−13 °C)–(−8 °C) |
| 27 | −18 | 28 | 11 | (−13 °C)–(−8 °C) | 57 | 6 | −24 | 9 | (1 °C)–(6 °C) |
| 28 | −33 | 21 | 6 | (1 °C)–(6 °C) | 58 | 17 | 1 | 7 | (−13 °C)–(−8 °C) |
| 29 | 16 | −37 | 22 | (−13 °C)–(−8 °C) | 59 | −37 | 40 | 11 | (1 °C)–(6 °C) |
| 30 | −40 | 6 | 18 | (−5 °C)–(0 °C) | 60 | −14 | 38 | 19 | (1 °C)–(6 °C) |

*4.2. Example Solution and Results Analysis*

4.2.1. Temperature Sensitivity Analysis

Fresh commodities have different sensitivity coefficients at different temperatures, which leads to differences in the value loss. Using the example R108, the temperature control cost and value loss under the temperature range (1 °C)–(6 °C), (−5 °C)–(0 °C) and (−13 °C)–(−8 °C) were calculated. The results are shown in Figure 7.

Figure 7 shows that the temperature control cost and value loss both change with temperature, but they are both paradoxical. When the temperature is at the left end of the interval, the value loss is the smallest and the temperature control cost is the largest. Conversely, when the temperature is the right endpoint of the interval, the commodity is at a higher temperature. The loss is the largest and the temperature control cost is the smallest. This indicates that under a constant external temperature, the value loss gradually increases and the temperature control cost gradually decreases with increasing temperature. The value loss and temperature control costs were different in the increase or decrease during the temperature change. Based on the Pareto optimal principle [41], the optimal temperature in the temperature control intervals (1 °C)–(6 °C), (−5 °C)–(0 °C) and (−13 °C)–(8 °C) are 3 °C, −3 °C and −11 °C, respectively. When the temperature is at its optimum, adjusting the temperature up or down does not result in an optimum.

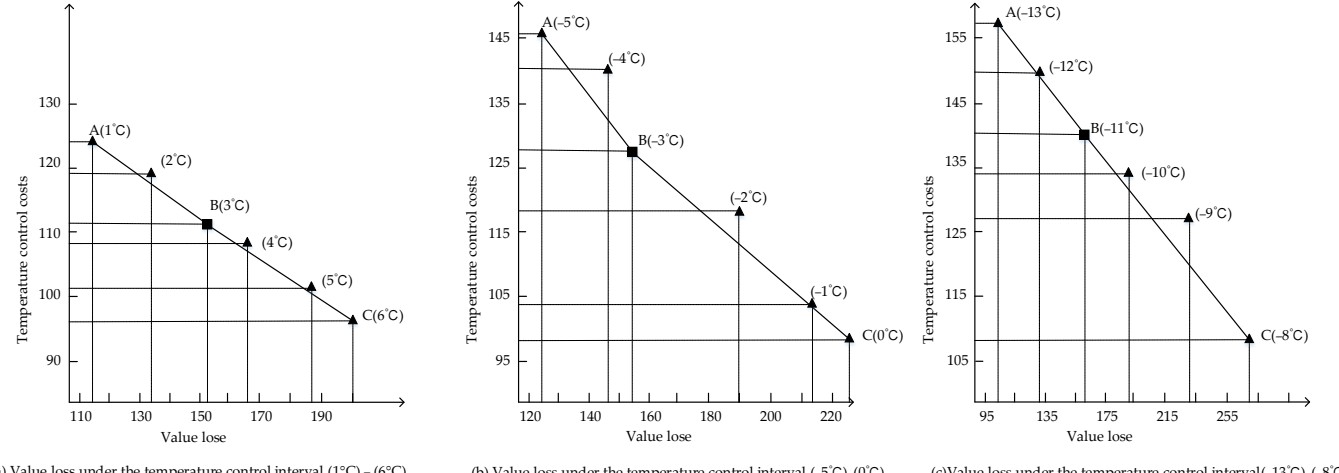

(a) Value loss under the temperature control interval (1°C) – (6°C)  (b) Value loss under the temperature control interval (–5°C)–(0°C)  (c)Value loss under the temperature control interval(–13°C)–(–8°C)

**Figure 7.** Fresh commodity value loss and temperature control paradox diagram.

### 4.2.2. Carbon Emissions Impact on GCVRPDD

To research the impact of carbon emission cost on the dynamic vehicle routing problem, this paper uses the AH-NOA to solve the dynamic vehicle routing problem, considering carbon emissions and not considering carbon emissions, respectively. The algorithm parameters used are the same. The relevant parameters are $C_2 = 5.2$ CNY/km and $f_{kw} = 0.12$ L/km. The algorithm was programmed using Matlab R2016a and implemented running on Windows 10, 8 G and 2.80 GHz.

(1) Results and Analysis of the Initial Scheme

(a)  Results of Initial Scheme

The AH-NOA was used to solve the vehicle path problem with and without considering carbon emissions. The initial distribution path is shown in Figure 8. Figure 8a shows that eight vehicles were dispatched. Figure 8b shows that nine vehicles were dispatched. There are some differences in the vehicle paths for the two distribution schemes, which are mainly due to whether or not carbon costs are taken into account. The specific routes for the two distribution schemes are shown in Tables 6 and 7, respectively. The stars in Figure 8 represent distribution centers and the dots represent customer points.

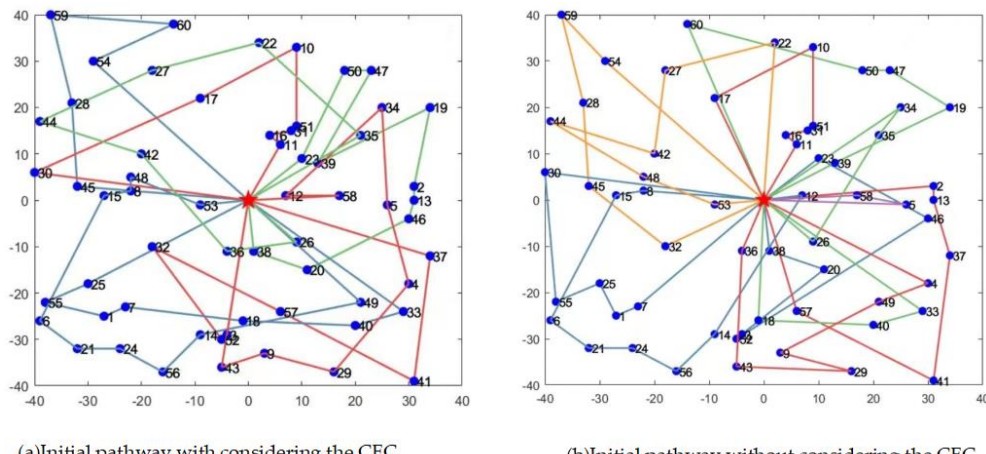

(a)Initial pathway with considering the CEC  (b)Initial pathway without considering the CEC

**Figure 8.** Vehicle routing.

**Table 7.** Initial route considering the carbon emission costs.

| Vehicle Serial No. | Pathway | Temperature Control Interval |
|---|---|---|
| 1 | 0−>45−>28−>59−>60−>54−>0 | (1 °C)–(6 °C) |
| 2 | 0−>3−>52−>32−>57−>41−>37−>0 | |
| 3 | 0−>23−>50−>47−>39−>19−>2−>13−>46−>20−>38−>0 | |
| 4 | 0−>33−>40−>18−>7−>1−>55−>25−>0 | (−5 °C)–(0 °C) |
| 5 | 0−>11−>16−>31−>51−>10−>17−>30−>0 | |
| 6 | 0−>49−>14−>56−>24−>21−>6−>15−>8−>48−>53−>0 | (−13 °C)–(−8 °C) |
| 7 | 0−>43−>9−>29−>4−>5−>34−>12−>58−>0 | |
| 8 | 0−>26−>36−>42−>44−>27−>22−>35−>0 | |

(b)　　Results Comparison

Comparing Tables 7 and 8, it can be seen that there are significant differences in the paths of the two schemes. There is a significant difference in the order of delivery customers. The carbon emission cost, value loss and total cost have obvious changes, as shown in Table 9. As can be seen from Table 9, the distribution scheme, considering the carbon emission cost, is CNY 119 more in value loss than without considering. However, the carbon emission cost decreased by 9.9%, while the transportation cost and total cost also decreased by 9.7% and 10.1%, respectively. This is largely attributable that the distribution scheme, considering carbon emission costs, needs to provide priority delivery to customers with larger loads. Although this causes an increase in value loss, it achieves a lower carbon emission cost and total cost during the shelf life of the commodity. Therefore, distribution schemes considering the carbon emission costs must reduce carbon emissions while minimizing the total cost.

**Table 8.** Initial route without considering the carbon emission costs.

| Vehicle Serial No. | Pathway | Temperature Control Interval |
|---|---|---|
| 1 | 0−>38−>20−>3−>52−>46−>23−>0 | (1 °C)–(6 °C) |
| 2 | 0−>2−>13−>37−>41−>57−>0 | |
| 3 | 0−>39−>19−>47−>50−>60−>0 | |
| 4 | 0−>32−>45−>28−>59−>54−>0 | |
| 5 | 0−>7−>1−>25−>55−>30−>0 | (−5 °C)–(0 °C) |
| 6 | 0−>11−>16−>31−>51−>10−>17−>0 | |
| 7 | 0−>33−>40−>18−>0 | |
| 8 | 0−>8−>15−>6−>21−>24−>56−>14−>12−>0 | (−13 °C)–(−8 °C) |
| 9 | 0−>36−>43−>29−>9−>49−>4−>0 | |

**Table 9.** Cost comparison of two distribution schemes.

| Distribution Scheme | VN | CEC | VL | TrC | TC |
|---|---|---|---|---|---|
| Considering the carbon emission costs | 8 | 599 | 318 | 370 | 2535 |
| Without considering the carbon emission costs | 9 | 665 | 237 | 410 | 2821 |

(2) Adjustment and Comparison Analysis of Pathway under Real-time Information

During the vehicle operation, the dispatch center receives information on all dynamic events, as shown in Table 10. To simplify the calculation, it is assumed that dynamic events only occur within a certain interval.

Dynamic events occur at [8:45, 9:00]. According to the dynamic adjustment strategy, the current dynamic event is adjusted in the next time slice based on the rolling time domain division. Thus, the dynamic event handling time slice is at [9:15, 9:30]. Based on the updated demand information, the AH-NOA is used to solve the GCVRPDD separately. The distribution route scheme is shown in Figure 9. As can be seen in Figure 9, vehicles

complete their tasks according to the new route arrangement. The specific routes are shown in Table 11.

**Table 10.** Dynamic event information.

| No. | Customer No. | X/km | Y/km | Dynamic Time | Dynamic Event | Temperature Control Interval |
|-----|-----|-----|-----|-----|-----|-----|
| 1 | 61 | 27 | −2 | 8:46 | New customer point demand 19 kg | (1 °C)–(6 °C) |
| 2 | 62 | −22 | −15 | 8:49 | New customer point demand 18 kg | (−13 °C)–(−8 °C) |
| 3 | 63 | −22 | −14 | 8:50 | New customer point demand 10 kg | (−13 °C)–(−8 °C) |
| 4 | 50 | 18 | 28 | 8:52 | Demand reduced by 10 kg | (1 °C)–(6 °C) |
| 5 | 28 | −33 | 21 | 8:58 | Demand increased by 5 kg | (1 °C)–(6 °C) |

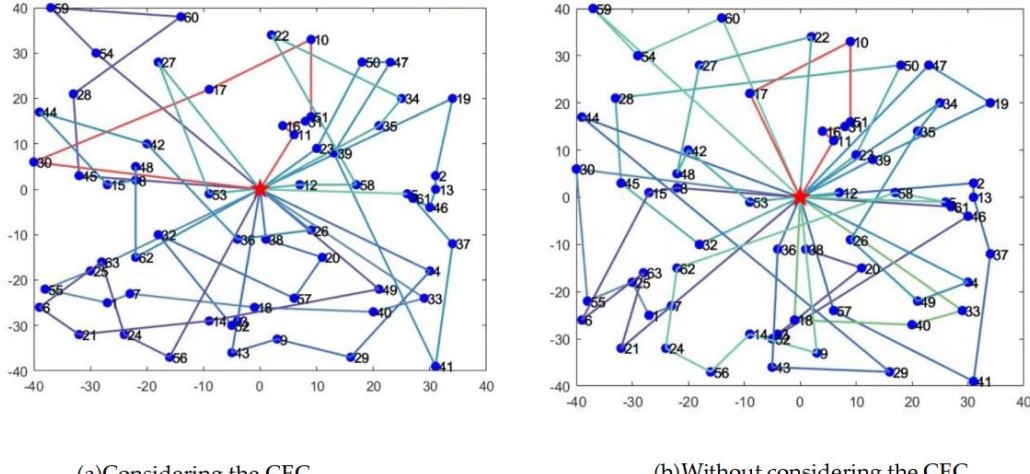

(a)Considering the CEC

(b)Without considering the CEC

**Figure 9.** Distribution paths after dynamic events.

**Table 11.** Specific distribution routes after change under the two distribution schemes.

| Distribution Scheme | Vehicle Serial No. | Pathway | Temperature Control Interval |
|-----|-----|-----|-----|
| Considering the carbon emission costs | 1 | 0−>28−>60−>59−>54−>0 | (1 °C)–(6 °C) |
| | 2 | 0−>3−>52−>32−>57−>20−>38−>0 | |
| | 3 | 0−>23−>50−>47−>39−>41−>37−>61−>46−>13 −>2−>19−>0 | |
| | 6 | 0−>49−>14−>21−>6−>63−>24−>56−>0 | (−13 °C)–(−8 °C) |
| | 7 | 0−>43−>9−>29−>4−>0 | |
| | 8 | 0−>26−>36−>42−>44−>15−>8−>48−>62−>0 | |
| | **9** | 0−>35−>34−>22−>58−>12−>53−>27−>0 | |
| | **10** | 0−>5−>0 | |
| Without considering the carbon emission costs | 1 | 0−>38−>20−>3−>52−>46−>61−>0 | (1 °C)–(6 °C) |
| | 2 | 0−>2−>13−>37−>41−>57−>0 | |
| | 3 | 0−>39−>19−>47−>0 | |
| | 4 | 0−>32−>45−>28−>50−>23−>0 | |
| | **13** | 0−>60−>54−>59−>0 | |
| | 8 | 0−>8−>15−>6−>63−>21−>0 | (−13 °C)–(−8 °C) |
| | 9 | 0−>36−>43−>29−>44−>0 | |
| | 10 | 0−>34−>35−>26−>49−>4−>0 | |
| | 11 | 0−>22−>27−>48−>42−>53−>12−>0 | |
| | 12 | 0−>5−>58−>62−>24−>56−>14−>9−>0 | |

Comparing Figure 9a,b, it can be seen that the two distribution schemes are completely different. There are also significant differences in the costs of each distribution. This is shown in Table 12. It can be seen from Table 12 that the distribution scheme considering carbon emissions increased in value loss by CNY 105 and decreased the transportation cost and total cost by 6.4% and CNY 36, respectively. Although there is a significant value loss, this loss is still within the goods freshness and the distribution scheme considering carbon emissions is 6.3% lower in carbon emissions, which is good for the positive corporate social image and corporate future development. The examples used are universally applicable because they are randomly generated based on actual situations. The above comparison can demonstrate that considering carbon emissions in the dynamic vehicle path can reduce carbon emissions and total cost. The AH-NOA can effectively solve the GCVRPDD. By comparing the initial route and adjusted costs of the two schemes, it was found that the distribution scheme considering carbon emissions had a greater change in the number of vehicles, carbon emission costs, transport costs and total costs. This is shown in Table 13.

**Table 12.** Cost comparison of two distribution schemes after dynamic events.

| Distribution Scheme | *VN* | *CEC* | *VL* | TrC | *TC* |
|---|---|---|---|---|---|
| Considering the carbon emission costs | 10 | 667 | 320 | 412 | 2823 |
| Without considering the carbon emission costs | 13 | 712 | 215 | 440 | 2859 |

**Table 13.** The extent of cost changes before and after distribution program adjustments.

| Distribution Scheme | Increased Number of Vehicles | Increased *CEC* | Increased TrC | Increased *TC* |
|---|---|---|---|---|
| Considering the carbon emission costs | 2 | 68 | 42 | 288 |
| Without considering the carbon emission costs | 1 | 47 | 30 | 38 |

## 5. Conclusions

This paper researches GCVRPDD based on the AH-NOA. Firstly, through the analysis of the problem, using virtual customer points to transform dynamic vehicle paths into SVRP based on the rolling time domain, and construct a two-stage GCVRPDD model with the goals of minimizing vehicle dispatch costs, transport costs, temperature control costs, value loss and carbon emissions costs. Secondly, the AH-NOA is designed according to the model characteristics and improvements are made to reduce its disadvantages. The CW algorithm is added to initialize partial solutions to ensure population diversity. Better nutcrackers are selected to ensure the overall quality of the population by generating new nutcrackers. The decay function is introduced to improve the convergence speed of the algorithm and to avoid falling into a local optimum. The algorithm is compared with the original NOA to verify the effectiveness and reasonableness of the algorithm improvements. According to the algorithm and the model, it found the optimal temperature in three different temperature control intervals. Finally, the GCVRPDD is solved based on two different distribution schemes. The results show that the distribution scheme considering carbon emission costs, while higher in value loss, is lower in carbon emissions and total costs and has a significantly fewer number of vehicles. Considering the carbon emission cost in DVRP can lead to a significant increase in carbon emissions, transport costs and total costs compared to SVRP. This is because the quality of fresh agricultural products declines as the vehicle time in transit increases. To reduce the value loss of fresh agricultural products, companies need to generate more carbon emissions to meet customer demand, resulting in increased carbon emissions costs, transportation costs and total costs. However, with the emergence of diversity in consumer demand, it is more realistic to consider carbon emissions in dynamic delivery. At the same time, the AH-NOA adds a new solution method to dynamic vehicle path optimization.

Future research will focus on the effects of distribution vehicle temperature dynamics, customer time windows, customer satisfaction and other factors to make the problem more realistic.

**Author Contributions:** D.W.: Supervision, Methodology and Writing—original draft. R.Y.: Formal analysis, Validation, Software and editing. H.J.: Conceptualization, Funding acquisition. F.C.: Resources. All authors have read and agreed to the published version of the manuscript.

**Funding:** This research was funded by the China National Social Science Fund Project "Study on dynamic optimization of urban main and non-staple food reserve and supply system under abnormal conditions" (No. 22BGL274), and the Major Projects of the National Social Science Foundation of China (No. 22ZDA058).

**Data Availability Statement:** All experimental data in this paper comes from https://people.idsia. ch//~luca/macs-vrptw/problems/welcome.htm (accessed on 10 March 2023.)

**Conflicts of Interest:** The authors declare no conflict of interest.

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
