# Peer review of "An Adaptive Nutcracker Optimization Approach for Distribution of Fresh Agricultural Products with Dynamic Demands"

_agriculture, doi:10.3390/agriculture13071430_

Round 1

Reviewer 1 Report

The topic of this research is interesting; however, the main weakness of this manuscript lies in the lack of explanation, particularly in Sections 1 to 3. 

To enhance the clarity and overall quality of the manuscript, some recommendation are addressed as follows:

1. It is essential to clarify the notation used for indices, variables, and parameters in the construction of the model.

2. It is necessary to provide more detailed explanation of the MEET model referred to in Eq. (1).

Furthermore, I was unable to locate the reference paper you cited as "[25] Zhou, X, C.; Liu, C, S.; Zhou, K, J.; He, C, H.; Huang, X, B,. Improved ant colony algorithm and modelling of time-dependent green vehicle routing problem. Journal of Management Sciences in China.2019, 22(05):57-68." 

I kindly request you to verify the accuracy of the reference or provide the link to the paper in your response to this reviewer's letter.

3. In Line 135, could you please explain the purpose of the thermodynamic temperature T(K)? Additionally, please clarification on how to calculate the temperature control cost factor mentioned in Line 140.

4. The concept of the dynamic adjustment strategy depicted in Figure 1 does not appear to correlate with the explanation provided in Section 2.5.3. It is recommend aligning the content in these sections to ensure consistency.

5. To better evaluate the superiority or advantages of the proposed approach, it would be beneficial to compare it with existing methods or algorithms. This will facilitate a more comprehensive assessment of its performance.

6. The significance of the experimental results should be determined through statistical testing. 

7. The determination of the significance of the experimental results should be based on statistical testing. In addition, it would be advantageous to provide a comprehensive discussion regarding the time complexity of the proposed method, taking into account its integration of multiple hybrid methods and searches.

8. Please provide a more thorough explanation of the K-means clustering method utilized in this research. Furthermore, it would be beneficial to highlight the significance and relevance of this method in the context of this study.

In addition to the major comments, we have identified a few minor issues:

1. Please note that in-text citations should not include full stops. For reference, you can refer to the guidelines at https://www.mdpi.com/journal/agriculture/instructions.

2. The NOA flow chart (Figure 3) would benefit from being presented in a standard top-down style.

Reviewer 2 Report

This manuscript addresses the problem of distributing fresh agricultural products to a set of customers located at different geographical locations. The distribution is done from a central distribution center with a fleet of homogeneous trucks having temperature control systems. The manuscript considers a situation in which customers’ orders can be dynamically received during the delivery process. In the considered problem, it is required to plan for the routes that the trucks will follow to fulfill the requested orders, while the total cost is minimized. The total cost is composed of 5 cost elements that include the truck dispatching cost, the transportation cost resulting from moving the trucks through the delivery routes, the temperature control cost, the cost of losses incurred during the distribution, and the carbon emission cost. The studied problem is referred to as the green-capacitated vehicle routing problem with dynamic demand (GCVRPDD). The manuscript proposes a solution approach based on an adaptive hybrid nutcracker optimizer algorithm (AH-NOA). Some computational experiments are conducted to compare the results obtained by the proposed solution approach with a plain nutcracker optimizer algorithm (NOA).  Results demonstrate that the proposed hybrid approach produces better results. Furthermore, a numerical example is presented to demonstrate the effects of temperature control and carbon emission considerations on the delivery routes.

In general, this manuscript addresses an interesting problem, however, it has some critical issues that need to be addressed as explained in the following points:
1)    The literature review is lacking prior recent contributions for solving closely related capacitated vehicle routing problems (VRPs). For the case of VRP with time windows (for which the test instances in section 3.2.4 are used), the literature has several recent contributions for implementing different solution techniques which are ignored by this manuscript.  
2)    The manuscript lacks a clear and precise description of the studied problem. Accordingly, section 2 needs to be rewritten to provide sufficient details about the structure of the studied problem. It is advisable to provide a table containing full descriptions of the used symbols in the model. It is also important to provide appropriate measuring units for all defined parameters. Furthermore, the presented model must be consistent with the numerical results presented in sections 3.2.4 and 4. For instance, the model defines the set of trucks (vehicles) as a given parameter K, meanwhile in Table 3, the “vehicle count” appears as a decision variable. Such inconsistencies must be resolved.
3)    The manuscript does not provide any justification for selecting the nutcracker optimizer algorithm for solving the studied problem. There are several solution approaches that implemented other metaheuristics with competitive results, so why did the authors choose such a metaheuristic with apparent shortcomings as stated in lines 280 and 281?
4)    The results presented in Table 3 are not sufficient to draw solid conclusions about the performance of the proposed AH-NOA approach. The authors should consider other metaheuristics from the literature that are successfully applied to a closely related VRP with time windows problem.

Beside the above major concerns, there are some other minor concerns that should be considered for improving the quality of the manuscript, they include:
5)    The language of the manuscript needs to be revised carefully as many typos are spotted.
6)    The reference citation style is not consistent with the MDPI template.
7)    Refer to your manuscript as “this paper” instead of “the paper” which appears in line 14, and “this study” instead of “the study” which appears in line 91.
8)    State clearly your contributions in the introduction section.
9)    All mathematical notations within text need to be carefully edited for consistency in size and spacing.
10)    In section 2.5, provide more explanation of how the developed framework works.
11)    In section 3.2.2, provide a reference for the nutcracker optimizer algorithm.
12)    Remove non-English characters such as the one in equation (45)
13)    In section 3.2.3, explain how the position vector is related to the decision variables of the model presented in section 2.5.3

In conclusion, this manuscript is not recommended for publication in its current form. A major revision is needed to address the above concerns.

The language of the manuscript needs to be revised carefully as many typos are spotted.

Round 2

Reviewer 1 Report

Thank you for the detailed revision. 

There are some minor suggestions, please check the attached file.

Reviewer 2 Report

The modifications are satisfactory.

Author Response

Thank you for your decision and constructive comments on my manuscript.